# Overview of Maximum Power Point Tracking Methods for PV System in Micro Grid

**Jae-Sub Ko [1], Jun-Ho Huh [2],\* and Jong-Chan Kim [3],\***

[1]  Department of Electrical Engineering, Sunchon National University, 255 Jungang-ro, Suncheon-city Jeollanam do 57922, Korea; kokos22@sunchon.ac.kr
[2]  Department of Data Informatics, Korea Maritime and Ocean University, Busan 49112, Korea
[3]  Department of Computer Engineering, Sunchon National University, 255 Jungang-ro, Suncheon-city Jeollanam do 57922, Korea
\*  Correspondence: 72networks@kmou.ac.kr (J.-H.H.); seaghost@sunchon.ac.kr (J.-C.K.); Tel.: +82-51-410-4347 (J.-H.H.); +82-61-750-3620 (J.-C.K.)

**Abstract:** This paper presents an overview of the maximum power point tracking (MPPT) methods for photovoltaic (PV) systems used in the Micro Grids of PV systems. In the PV system, the output varies nonlinearly with temperature and radiation, and the point at which power is maximized appears accordingly. The MPPT of the PV system can improve output by about 25%, and it is very important to operate at this point at all times. Various methods of tracking the MPP of the PV system have been studied and proposed. In this paper, we discuss commonly used methods for the MPPT of PV systems, methods using artificial intelligence control, and mixed methods, and present the characteristics, advantages, and disadvantages of each method.

**Keywords:** PV system; MPPT; artificial intelligent; Micro Grid

## 1. Introduction

The photovoltaic (PV) system is receiving much attention because it is an infinite, eco-friendly energy source. In addition, since the PV system generates electricity without a driving unit, it has the advantage of a long life, as it requires little maintenance. Nonetheless, PV systems are highly dependent on environmental conditions, and they have the disadvantage of low conversion efficiency [1–5]. In addition, the PV system can only be generated at the time when there is sunlight, and its characteristics change nonlinearly according to the surrounding environment [5–7]. In particular, PV systems are highly dependent on temperature and radiation. Temperature affects the voltage of the PV system, and radiation affects current. The power of a PV system has a characteristic of changing nonlinearly with temperature and radiation, and there is a point where power is maximized under certain conditions. Therefore, the technique of controlling the PV system so that it can always be operated at this point is very important to improve the efficiency and output of the PV system. The method for tracking the maximum power point of the PV system requires "fast tracking speed in transient", "low vibration in steady state", "responsiveness to temperature and radiation changes", and "easy implementation". The maximum power point tracking (MPPT) of the PV system can improve the power produced by 25%, and various methods have been studied for this [8–12].

Typical methods for MPPT include the constant voltage (CV) method [13,14], the open circuit voltage (OCV) method [13–20], the short circuit current (SCC) [21–23] method, the perturbation and observation (P&O) method [24–28], and the Incremental Conductance (IncCond) method [29–35]. The CV method is a method using one fixed voltage obtained under specific conditions, and the OCV method is a method using a certain percentage of the open circuit voltage of the PV module

as a reference voltage. The SCC method is a method of tracking the maximum power point by using a certain ratio of the short-circuit current as a reference current. The P&O method is a method wherein voltage or current is perturbed and the power is observed to control the direction in which power increases, whereas the IncCond method tracks the maximum power point by comparing the instantaneous conductance and incremental conductance. In addition, methods for improving the maximum power point tracking performance using artificial intelligence control techniques such as fuzzy control [35–44] and neural networks were studied [45–62]. The Micro Grid is constructed by connecting small-scale power grids to each other, and can replace existing large-scale power generation systems using fossil fuels, and has the advantage of reducing transmission loss because it is produced in places where energy is required. Micro Grid consists of DC power grid and distributed power. The PV system is the most representative distributed power supply for Micro Grids, and MPPT control is used to stably operate multiple power management units through distributed power, and the representative MPPT methods, the P and O method [63–65], and IncCond method [66] are used.

This paper introduces various methods for the MPPT control of the PV system, which is receiving much attention as an alternative energy source and is preferred for constructing smart grids and Micro Grids. It explains the operation principle, structure, advantages, and disadvantages of various MPPT methods, and introduces methods to overcome the disadvantages of each method. The various MPPT methods introduced in this paper will help engineers and researchers using PV systems to select the appropriate MPPT method according to the type, location, and environmental conditions of the PV system. In addition, it is expected that various ideas will be provided to study methods for improving the conventional MPPT method through a method in which the existing MPPT method is mixed with each other. The rest of this paper is organized as follows: Chapter 2 deals with solar cell modeling; Chapter 3 discusses various methods for MPPT control; finally, Chapter 4 presents the conclusion.

## 2. Solar Cell Modeling

Solar cells consist of one ideal diode and a constant current source ($I_{ph}$); since it is impossible to make an ideal diode in reality, however, a series resistor ($R_s$) and a parallel resistor ($R_{ph}$) representing the contact resistance and sheet resistance of the surface layer must be considered. Part of the light incident on the surface of the solar cell is reflected from the surface, and the light transmitted through the surface is absorbed in the solar cell, with the number of photons decreasing exponentially. Figure 1 shows the equivalent circuit of a solar cell.

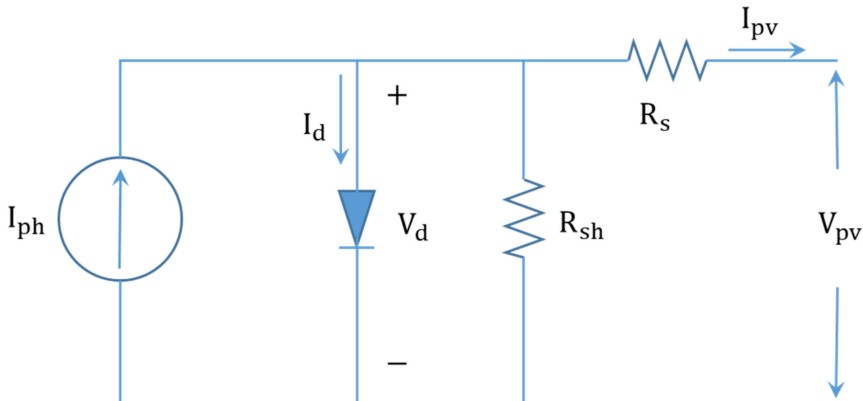

**Figure 1.** Equivalent of solar cell.

The photocurrent ($I_{ph}$) is proportional to the solar radiation and is given by the following equation:

$$I_{ph} = \left(\frac{G}{G_0}\right)I_{g0} + J_0(T_c - T_{ref}) \tag{1}$$

where $G_0$ is the reference solar irradiance, $I_{g0}$ is the current at the reference solar irradiance ($G_0$), $J_o$ is the temperature coefficient for photocurrent ($I_{ph}$), $T_c$ is the temperature of the cell, and $T_{ref}$ is the reference temperature of the cell. Most of the reference radiation used in the equation representing photocurrent is 1000 [W/m²]. The diode current ($I_d$) is given by the Shockley equation in Figure 1.

$$I_d = I_0\left[e^{\left(\frac{q(V_{pv}+I_{pv}R_s)}{nkT_c}\right)} - 1\right] \tag{2}$$

Here, $V_{pv} + I_{pv}R_s$ represents the voltage of the diode, $I_0$ denotes the diode inverse saturation current, and q represents the amount of electrons [$q = 1.602 \times 10^{-19}$ [c]. $V_{pv}$ and $I_{pv}$ are the cell voltage and current, respectively, $R_s$ is the series resistance, n is the ideal coefficient, and k is the Boltzmann constant ($k = 1.38 \times 10^{-23}$ [J/K]). The diode reverse saturation current $I_0$ is temperature-sensitive and can be expressed as:

$$I_0 = I_{d0}\left(\frac{T_c}{T_{ref}}\right)^3 e^{\left[\frac{qE_g}{nk}\left(\frac{1}{T_{ref}} - \frac{1}{T_c}\right)\right]} \tag{3}$$

where $I_{d0}$ represents the diode inverse saturation current at the reference temperature and $T_{ref}$ and $T_c$ use the Kelvin temperature. Bandgap energy $E_g$ of the silicon semiconductor constituting the solar cell can be expressed by the following equation:

$$E_g = 1.17 - \frac{4.73 \times 10^{-4} \times T_c^2}{T_c + 636} \tag{4}$$

The temperature of the solar cell ($T_c$) is proportional to the amount of solar radiation and can be expressed as:

$$T_c = 273 + T_a + \left(\frac{NOCT - 20}{800}\right) \times G \tag{5}$$

where $T_a$ represents the atmospheric temperature (°C) and NOCT (Nominal Operating Cell Temperature) denotes the nominal solar cell operating temperature. The relation of $I_{pv} - V_{pv}$ in the equivalent circuit of Figure 1 can be expressed as follows:

$$I_{pv} = I_{ph} - I_0\left[e^{\frac{q(V_{pv}+I_{pv}\times R_s)}{nkT_c}} - 1\right] - \frac{V_{pv} + I_{pv}R_s}{R_{sh}} \tag{6}$$

where $R_{sh}$ represents the parallel resistance. The current in Equation (6) is common to the left and right equations, and the relation of $I_{pv} - V_{pv}$ can be expressed as follows [67–73]:

$$f\left(I_{pv}, V_{pv}, G\right) = I_{pv} - \left\{I_{ph}(G) - I_0(G)\left[e^{\left(\frac{q(V_{pv}+I_{pv}R_s)}{nkT_c(G)}\right)} - 1\right] - \left(V_{pv} + \frac{I_{pv}R_s}{R_{sh}}\right)\right\} = 0 \tag{7}$$

## 3. MPPT Methods

### 3.1. Constant Voltage Method

As the simplest among various methods for the MPPT control of a PV system [13,14], the CV method is a method of using a reference voltage ($V_{ref}$) for the maximum power point voltage ($V_{mpp}$) obtained under standard test conditions (STC: radiation 1000 W/m², cell temperature 25 °C, AM 1.5) of a PV module or a voltage determined under specific conditions. Therefore, no additional input is needed to calculate the reference voltage, and the voltage must be measured to control the voltage of the PV module as the reference voltage. The MPP of the PV module varies with temperature and solar radiation. Since the CV method uses a fixed reference voltage for specific radiation and temperature conditions, however, this method has a problem—i.e., it cannot accurately track the MPP.

Figure 2 shows the flowchart for the CV method. Measure the current PV module voltage and compare it with the reference voltage to change the duty ratio in order to track the maximum power point.

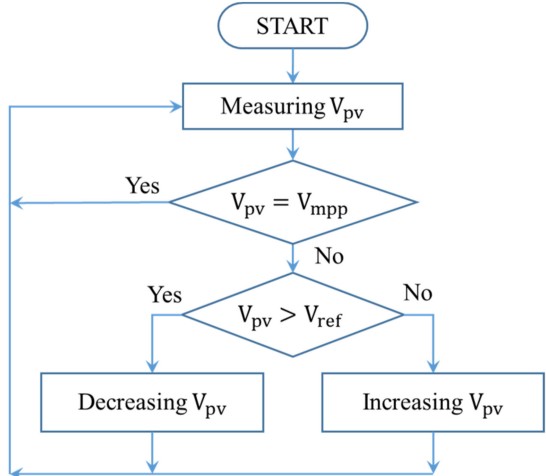

**Figure 2.** Flowchart of CV method.

### 3.2. Open Circuit Voltage (OCV) Method

The OCV method is a method of compensating for the shortcomings of the CV method. Since the CV method is controlled only at a constant voltage regardless of radiation and temperature, it has a disadvantage of not operating at an accurate maximum power point. Figure 3 shows the location of the open circuit voltage ($V_{oc}$) and maximum power point (MPP) voltage ($V_{mpp}$) when rated power of PV module is 110 Wp (Watt peak) and 30 Wp.

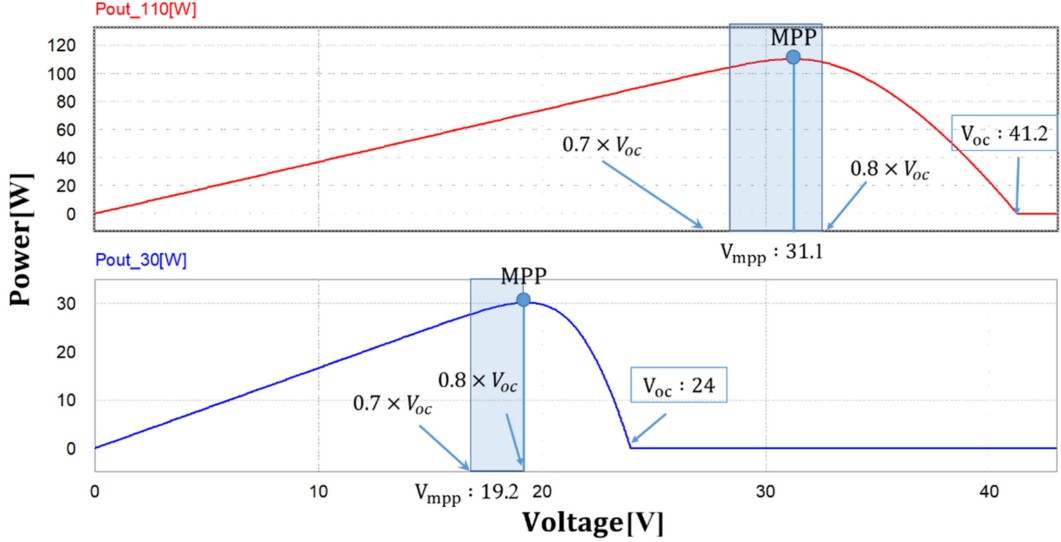

**Figure 3.** Location of $V_{oc}$ and $V_{mpp}$.

The OCV method calculates the reference voltage for tracking the maximum power point using Equation (8), based on the characteristic that the voltage at the maximum power point is generally present between 70% and 80% of the open circuit voltage.

$$V_{mpp} = V_{oc} \times k_1 \quad 0.7 \leq k_1 \leq 0.8 \tag{8}$$

The advantage of OCV lies in its simple implementation. Since the open circuit is configured by separating the load in order to measure the open circuit voltage according to constant cycle or condition, however, it has the disadvantages of non-continuous power supply and occurrence of loss due to periodic separation and connection. In addition, since the location of the maximum power point is approximated, In Figure 3, the maximum power point is included in the range of Equation (8), but depending on the value selected, the OCV method can deviate significantly from the actual maximum power point.

Figure 4 shows the flowchart of the OCV method. Configure the open circuit according to the condition or a constant period to measure the open circuit voltage and calculate the reference voltage. This is a method of tracking the maximum power point by comparing the calculated reference voltage with the current PV voltage.

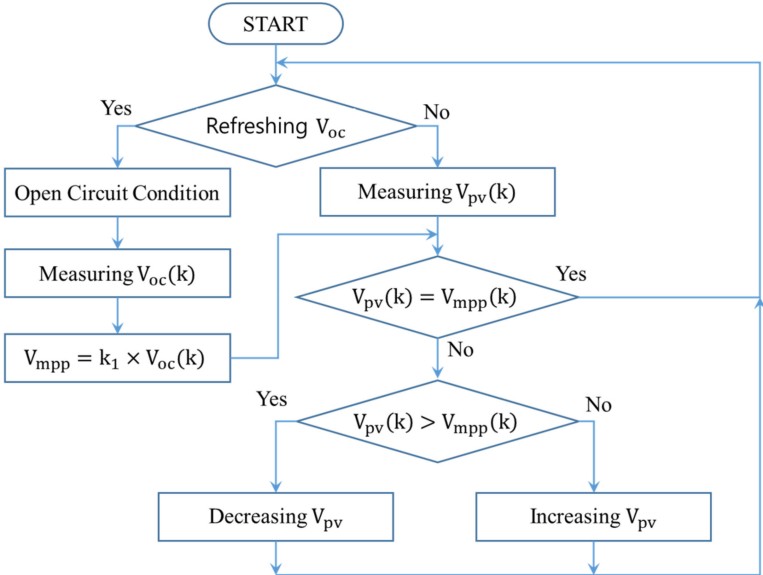

**Figure 4.** Flowchart of OCV Method.

### 3.3. Short Circuit Current (SCC) Method

The SCC method uses the characteristic of the maximum power current ($I_{mpp}$) falling within a certain range of the short circuit current ($I_{sc}$), which is the maximum current of the PV module. Equation (9) represents the current ($I_{mpp}$) that becomes the maximum power. The $k_2$ value is approximated by experiment, and a value between approximately 0.78 and 0.92 is used.

$$I_{mpp} = I_{sc} \times k_2 \; 0.78 \leq k_2 \leq 0.92 \tag{9}$$

Figure 5 shows the power-current (P-I) curve of the PV module. The current ($I_{mpp}$) that becomes the maximum power point is about 90% of the short-circuit current ($I_{sc}$).

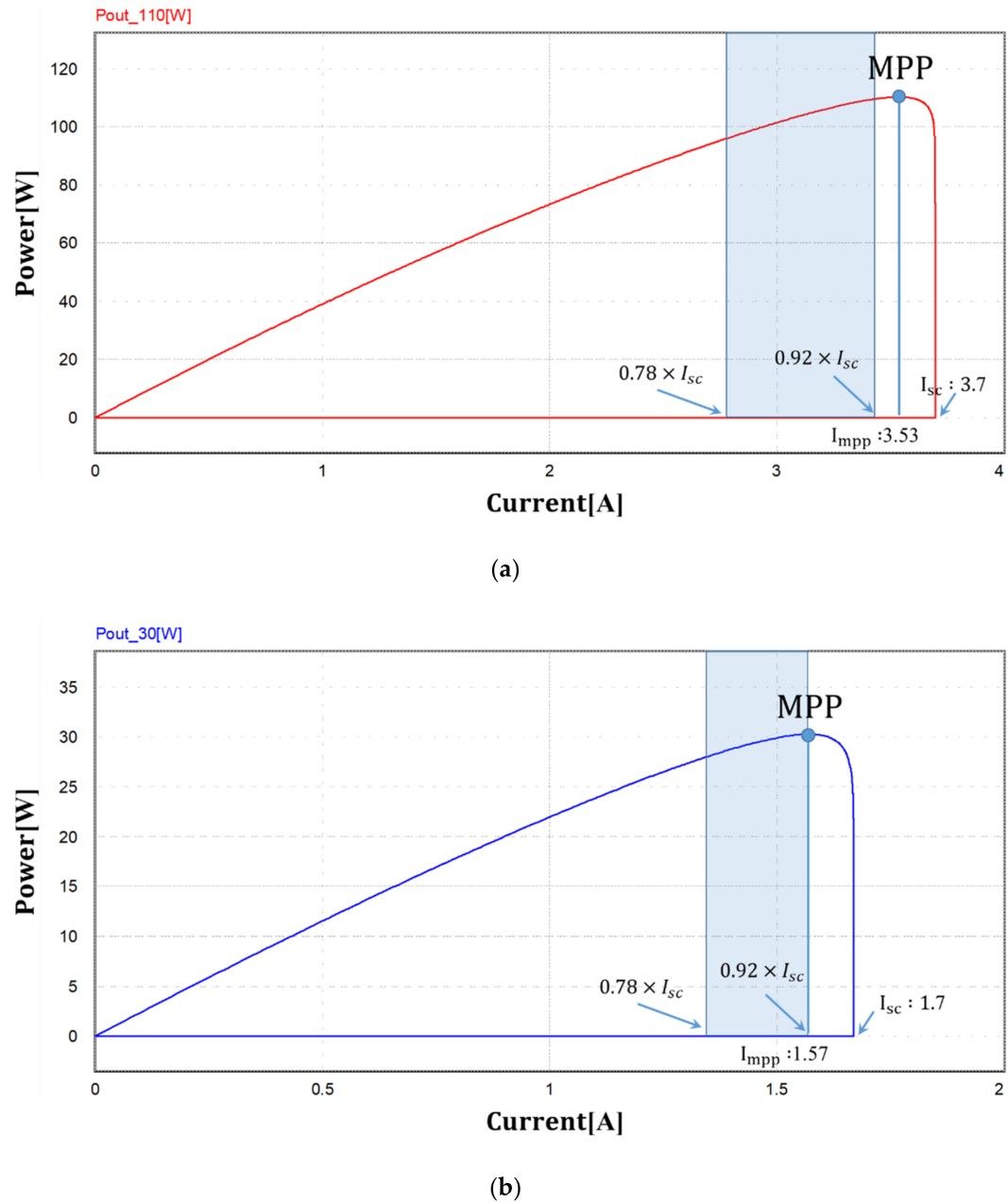

**Figure 5.** P-I curve of PV module. (**a**) PV module: 110 Wp; (**b**) PV module: 30 Wp.

Figure 6 shows the flowchart of the SCC method. The SCC method shorts the load to measure the short circuit current ($I_{sc}$) according to a certain period or condition. The reference current is calculated using the measured short-circuit current and the proportional constant $k_2$, compares it with the PV current ($I_{pv}$), and increases or decreases the PV current to track the maximum power point.

Like the OCV method, the SCC method requires periodic short-circuit current measurement; for this, the load must be shorted to form a short circuit. At this time, since current is not supplied to the load, power loss occurs, and efficiency is reduced, with the maximum power point current calculated by Equation (9) approximately calculated. Thus, there is a problem, since it is not an actual maximum power point.

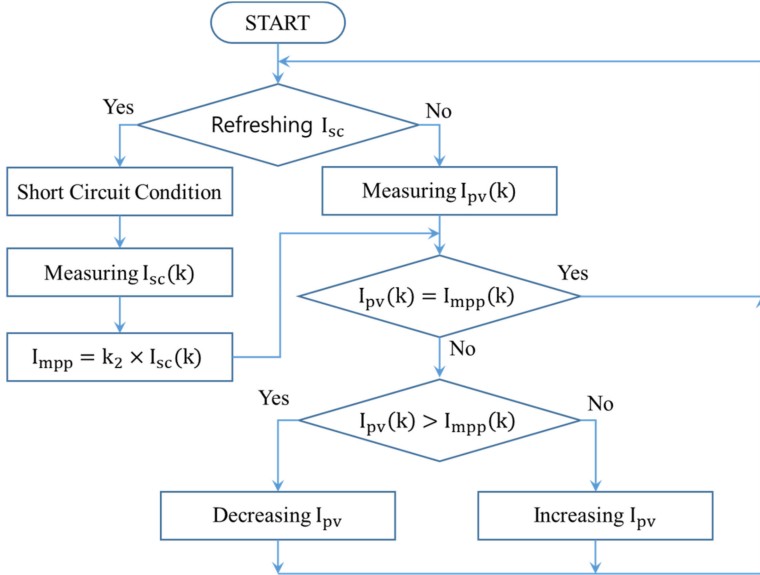

**Figure 6.** Flowchart of SCC method.

### 3.4. Perturbation and Observation (P&O) Method

The P&O method continuously controls the current or voltage in the direction of increasing power by perturbing the PV voltage or current and observing power. Table 1 shows the control direction of the next step by voltage or current perturbation and power observation.

**Table 1.** Direction of perturbation of P&O method.

| Perturbation of $P_{pv}$ or $I_{pv}$ | Observation of $P_{pv}$ | Direction of Perturbation |
| --- | --- | --- |
| Positive | Positive | Positive |
| Positive | Negative | Negative |
| Negative | Positive | Negative |
| Negative | Negative | Positive |

Figure 7 shows the flowchart of the P&O method. In the P&O method, PV module power is calculated by the measured voltage and current, with the maximum power point tracked by changing the reference value ($V_{mpp}$), which is the maximum power point according to the power and voltage change, by a certain size (changing value: $C_v$).

Since the P&O method is sequentially controlled according to the perturbation direction of the voltage or current and the direction of change of the observed power, the control algorithm is simple, and there is no open or short circuit of the load. Thus, it is possible to supply power to the load continuously. Since it continuously measures voltage and current, it has the advantage of being controlled according to the environmental conditions. When the radiation is low or rapidly changed, however, the exact maximum power point cannot be tracked, or vibration occurs near the maximum power point, thus resulting in loss. In order to solve this problem, a method of adjusting the changing value ($C_v$) of voltage or current for tracking the maximum power point has been proposed.

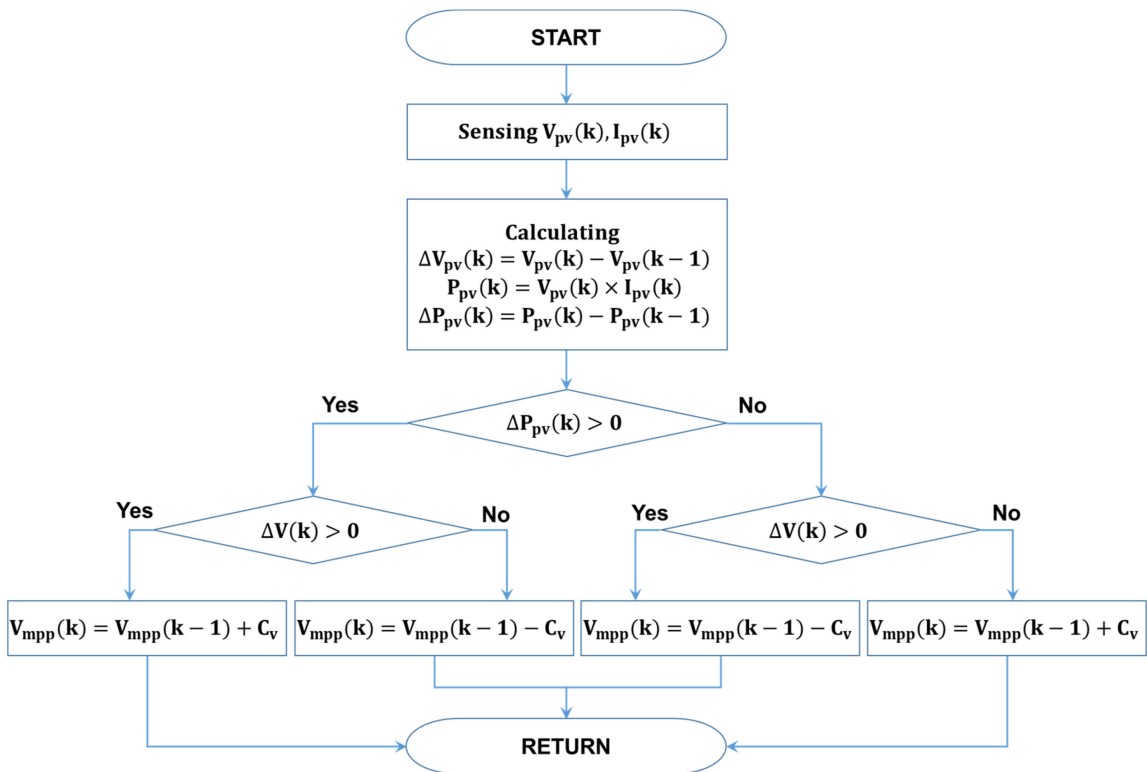

**Figure 7.** Flowchart of P&O method.

Equation (10) shows the equation for adjusting $C_v$ using the change in power, and Equation (11) presents the equation for adjusting $C_v$ using the slope of power in the P-V curve [74–77].

$$C_v = k_3 \times \left| \Delta P_{pv} \right| \tag{10}$$

$$C_v = k_3 \times \frac{\left| \Delta P_{pv} \right|}{\left| \Delta V_{pv} \right|} \tag{11}$$

Figure 8 shows the P-V curve of the PV module and the change in $C_v$ by Equation (11). Equations (12)–(14) show three zones according to the size of $C_v$.

$$\text{Zone 1}: \ k_3 \times \frac{\left| \Delta P_{pv} \right|}{\left| \Delta V_{pv} \right|} > K_{11} \tag{12}$$

$$\text{Zone 2}: K_{11} > k_3 \times \frac{\left| \Delta P_{pv} \right|}{\left| \Delta V_{pv} \right|} > K_{12} \tag{13}$$

$$\text{Zone 3}: K_{12} > k_3 \times \frac{\left| \Delta P_{pv} \right|}{\left| \Delta V_{pv} \right|} \tag{14}$$

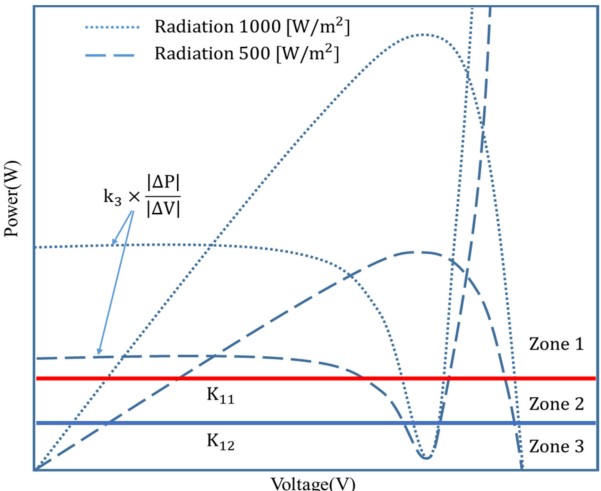

**Figure 8.** Changing value ($C_v$) and Zones 1~3 in the P-V curve.

Figure 9 shows how to set the $C_v$ value according to the three zones determined by Equations (12)–(14) and use it to track the maximum power point [78].

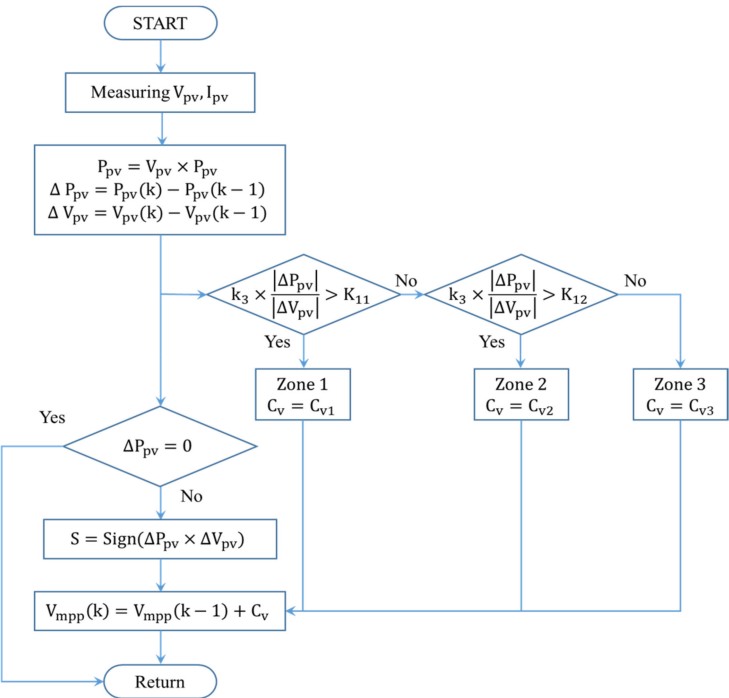

**Figure 9.** Flowchart of the modified P&O method.

Figure 10 shows the process of tracking the maximum power point in the P-V curve with the P&O method. In Figure 10, if the voltage and power changes are approximated in a triangular shape, the length of the hypotenuse can be approximated as in Equation (15). Since the change in power in the P-V curve decreases as it approaches the maximum power point, the hypotenuse length in Equation (15) decreases as it approaches the maximum power point. Using this characteristic, the amount of change for tracking the maximum power point can be varied as shown in Equation (16) [77].

$$\text{hypotenuse lenght} = \sqrt{\Delta P_{Pv}^2 + \Delta V_{pv}^2} \tag{15}$$

$$C_v = M \cdot \sqrt{\Delta P_{Pv}^2 + \Delta V_{pv}^2} \tag{16}$$

where M is a constant of change amount according to the parameter of the PV module.

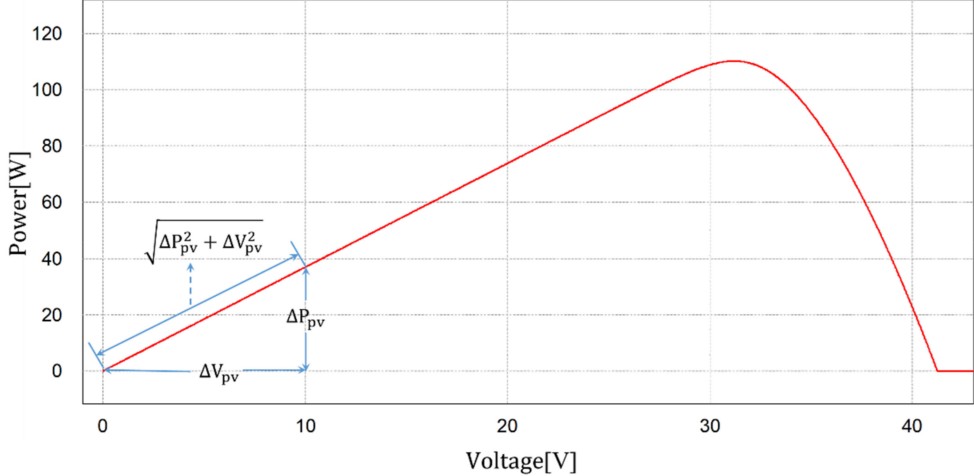

**Figure 10.** Changing voltage and power in the P-V curve.

Another method of varying the change value was proposed using fuzzy control. Figure 11 shows the method of adjusting the change value by using the slope ($S_p$) of power and the magnitude of the amount of change ($S_t$) as the input of fuzzy control in the P-V curve [79].

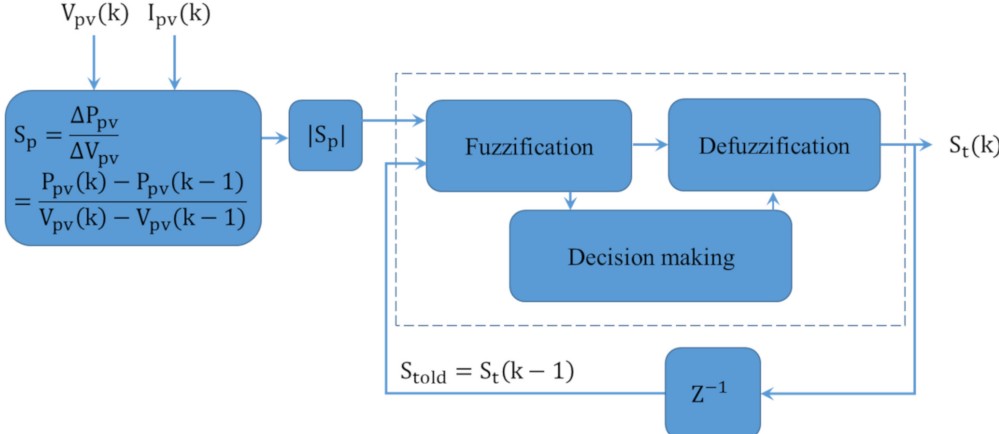

**Figure 11.** Variable step using Fuzzy control I.

Figure 12 shows the method of using the voltage and current of the PV module as the input of fuzzy control and calculating the change amount ($\Delta C_v$) of $C_v$ [80].

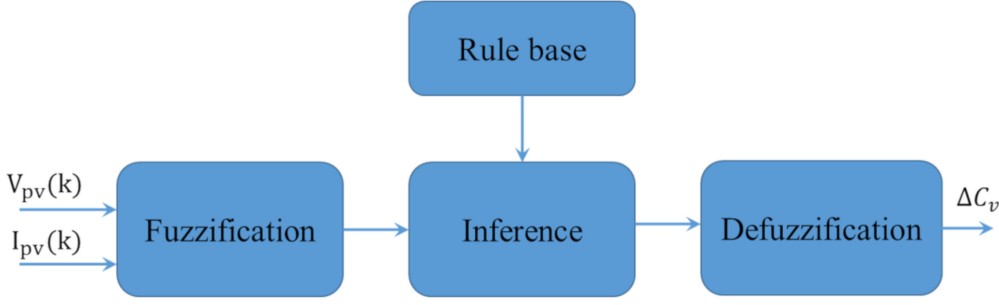

**Figure 12.** Variable step using Fuzzy control II.

### 3.5. Incremental Conductance (IncCond) Method

Under rapidly changing radiation conditions, the P&O method has a problem, i.e., it cannot track an accurate maximum power point. To improve this, the IncCond method was proposed.

The IncCond method uses a slope condition using a change in voltage and power in the P-V curve. Figure 13 shows the P-V curve and slope. The maximum power point position is determined by the magnitude of the slope. In Figure 13, point B with slope of 0 is the maximum power point, A with a negative slope is right of the maximum power point, and C with a positive slope is left of the maximum power point. In order to be in the direction of the maximum power point from the right side of the maximum power point where point A is located, the magnitude of the voltage must be reduced, and the left side of the maximum power point where point B is located must increase the voltage to move toward the maximum power point.

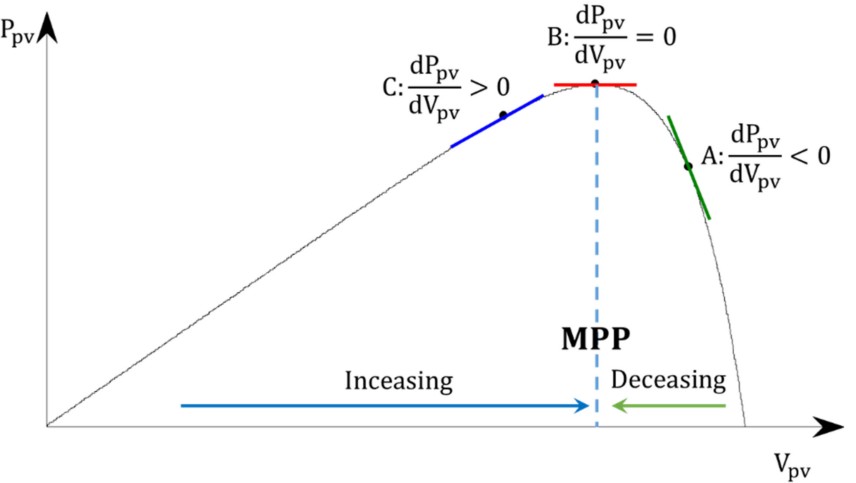

**Figure 13.** P-V curve and slope conditions.

Equations (17)–(20) show the equation for the slope of the P-V curve. The IncCond method refers to a method of controlling by comparing instantaneous conductance ($I_{pv}/V_{pv}$) and incremental conductance ($dI_{pv}/dV_{pv}$).

$$\frac{dP_{pv}}{dV_{pv}} = \frac{d(V_{pv}I_{pv})}{dV_{pv}} = \frac{dV_{pv} \times I_{pv}}{dV_{pv}} + \frac{V_{pv} \times dI_{pv}}{dV_{pv}} = I_{pv} + V_{Pv}\frac{dI_{pv}}{dV_{pv}} \tag{17}$$

$$A: \frac{dP_{pv}}{dV_{pv}} = I_{Pv} + V_{pv}\frac{dI_{pv}}{dV_{pv}} < 0 \rightarrow \frac{dI_{pv}}{dV_{pv}} < -\frac{I_{pv}}{V_{pv}} \tag{18}$$

$$B: \frac{dP_{pv}}{dV_{pv}} = I_{Pv} + V_{pv}\frac{dI_{pv}}{dV_{pv}} = 0 \rightarrow \frac{dI_{pv}}{dV_{pv}} = -\frac{I_{pv}}{V_{pv}} \tag{19}$$

$$C: \frac{dP_{pv}}{dV_{pv}} = I_{Pv} + V_{pv}\frac{dI_{pv}}{dV_{pv}} > 0 \rightarrow \frac{dI_{pv}}{dV_{pv}} > -\frac{I_{pv}}{V_{pv}} \tag{20}$$

Figure 14 shows the flowchart of the IncCond method. The voltage and current are measured using a sensor, and the change value is calculated using this. If the maximum power point is not reached, control is performed through Case A because the voltage changes continuously. Case A compares the incremental conductance ($\Delta I_{pv}/\Delta V_{pv}$) with the instantaneous conductance ($I_{pv}/V_{pv}$) to track the maximum power point according to its size.

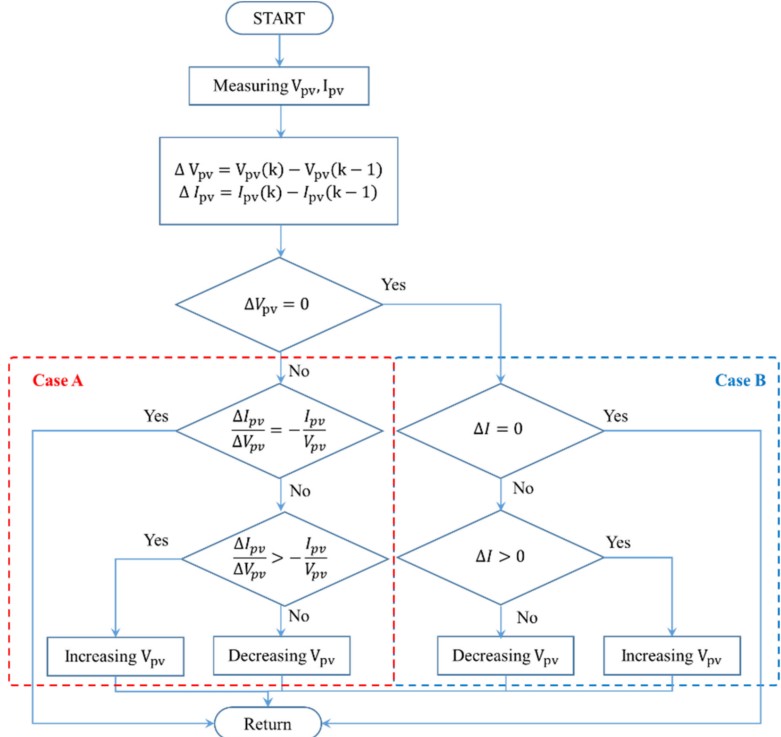

**Figure 14.** Flowchart of the IncCond method.

The P&O method is weak against rapidly changing radiation because there is no control method for radiation change during operation at the maximum power point. In order to overcome the disadvantages of this P&O method, the IncCond method detects the change in radiation by performing Case B in Figure 14 when the voltage does not change due to operation at the maximum power point. The two factors that affect the output of the PV module are temperature and radiation, with temperature affecting voltage and radiation affecting current. Therefore, Case B detects a change in radiation based on the change in current and increases the reference voltage ($V_{mpp}$) for maximum power point tracking when current increases ($\Delta I_{pv} > 0$); when current decreases, the reference voltage decreases conversely.

Generally, the value that changes the reference voltage for maximum power point control is used as a fixed value. This fixed size determines the maximum power point tracking speed and accuracy. A large changing value ($C_v$) can perform fast tracking but causes vibration around the MPP, which reduces tracking accuracy. The small changing value ($C_v$) increases accuracy by reducing the vibration near the MPP but has the disadvantage of slowing the tracking speed. Therefore, it is very important to set this change value properly.

In the IncCond method, methods for improving control performance by adjusting the changing value ($C_v$) have been proposed. A representative method is the method using the slope of the P-V curve as shown in Equations (10) and (11) [81,82].

*3.6. Fuzzy MPPT Method*

As one of the most widely used techniques for MPPT control, fuzzy control does not require accurate mathematical modeling and has the advantage of handling nonlinear systems. Fuzzy control generally consists of fuzzification, rule base, and defuzzification. Fuzzification converts numeric input into linguistic input variables based on the membership function shown in Figure 15. Defuzzification is the inverse process of fuzzification.

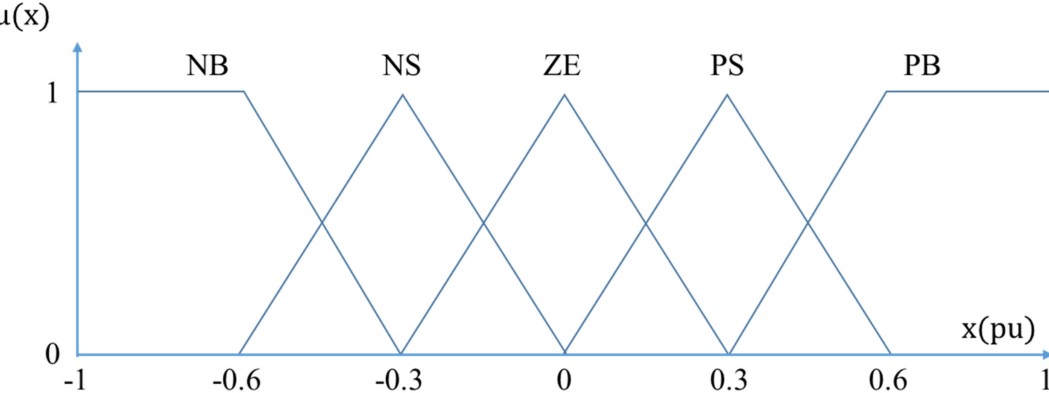

**Figure 15.** Membership function for inputs and output of fuzzy control.

In Figure 15, five language variables (NB: Negative Big, NS: Negative Small, ZE: Zero, PS: Positive Small, PB: Positive Big) are used. As the number of language variables increases, precision improves but the processing time of the algorithm increases.

Membership functions are formed using straight lines or curves. Curves are highly accurate but difficult to implement. The typical linear form is the trapezoidal and triangular membership functions, which have an advantage in real-time control; in fact, their various advantages are suggested in several studies. In particular, the triangular membership function is used in various ways because it has advantages in terms of response speed and steady-state error [83–87]. The triangular membership function generally uses a symmetrical shape as shown in Figure 15. If the importance of a specific section is high, however, the part can be modified and used [35,39,43,44,88].

Equations (21) and (22) show the error and changing value most frequently used as inputs in fuzzy control. Equation (21) is the slope of the P-V curve, and its size decreases as it approaches the maximum power point. Therefore, when this value becomes 0, it becomes the maximum power point; as such, this value is used as an input for fuzzy control. Equation (22) represents the changing error.

$$E(k) = \frac{\Delta P_{pv}}{\Delta V_{pv}} = \frac{P_{pv}(k) - P_{pv}(k-1)}{V_{pv}(k) - V_{pv}(k-1)} \text{ or } E(k) = \frac{I_{pv}}{V_{pv}} + \frac{\Delta I_{pv}}{\Delta V_{pv}} \tag{21}$$

$$\Delta E(k) = E(k) - E(k-1) \tag{22}$$

Table 2 shows the rule base of fuzzy control; this rule base is designed to control the duty ratio of the boost converter. The rule base of fuzzy control can be configured in various ways depending on the designer or user, and the control performance is highly dependent on the rule base.

**Table 2.** Rule base for fuzzy control.

| E \ ΔE | NB | NS | ZE | PS | PB |
|---|---|---|---|---|---|
| NB | ZE | ZE | NB | NB | NB |
| NS | ZE | ZE | NS | NS | NS |
| ZE | NS | ZE | ZE | ZE | PS |
| PS | PS | PS | PS | ZE | ZE |
| PB | PB | PB | PB | ZE | ZE |

Fuzzy control calculates the membership strength by the membership function of Figure 15 according to the size of the current error and the changing error, with the fuzzy inference calculating the control amount using Mamdani's Min-Max method.

The "IF THEN" rule for multiple inputs has "AND" and "OR" operations; the "AND" operation uses the Min rule, and the "OR" operation uses the Max rule.

If these rules are expressed as equations, Equations (23)–(26) are the resulting equations.

$$\text{IF x is } \mu(x_1) \text{ AND x is } \mu(x_2) \text{ AND } \cdots \text{x is } \mu(x_n) \text{ THEN y is Y} \tag{23}$$

$$\mu(x) = \text{Min}[\mu(x_1),\ \mu(x_2),\ \mu(x_3), \cdots, \mu(x_n)] \tag{24}$$

$$\text{IF x is } \mu(x_1) \text{ OR x is } \mu(x_2) \text{ OR } \cdots \text{x is } \mu(x_n) \text{ THEN y is Y} \tag{25}$$

$$\mu(x) = \text{Max}[\mu(x_1),\ \mu(x_2),\ \mu(x_3), \cdots, \mu(x_n)] \tag{26}$$

where x represents the input and $\mu(x_1)$ represents the strength of membership function. y is an output variable, and Y is an output value. The defuzzification of fuzzy control uses the center of gravity (COG) method the most and is denoted as follows:

$$\Delta y = \frac{\sum_{j=1}^{n} \mu(x)_j \times y_j}{\sum_{j=1}^{n} \mu(x)_j} \tag{27}$$

Fuzzy control does not have flowcharts such as P&O or IncCond, and it is strong in tracking speed and accuracy because it controls by changing the control amount according to the surrounding environment.

### 3.7. Neural Network Method

Neural networks have advantages such as robust operating, fast tracking, non-linear system processing power, and off-line training, and they are used in various ways for the MPPT of PV systems [45–62]. Nonetheless, the MPPT method using a neural network has a disadvantage of increased cost because a high-performance microcontroller is required compared to other methods. Neural networks are generally composed of three layers: input layer, hidden layer, and output layer. The number of nodes in each layer is used varyingly depending on the user who designs the controller. When MPPT control using a neural network is classified according to the type of input value, it can be divided into a method of using an electrical signal, a method of not using an electrical signal, and a method of mixing the two signals. The electrical signal input is the same as the voltage and current of the PV module, and the non-electrical input is the same as the temperature and radiation. The output includes reference power ($P_{pv}^*$), reference voltage ($V_{pv}^*$) and reference current ($I_{pv}^*$) for tracking the maximum power point of the PV system; by using this, the maximum power point is adjusted by adjusting the duty ratio of the power converter to track. Figure 16 shows the type of neural network classified according to input.

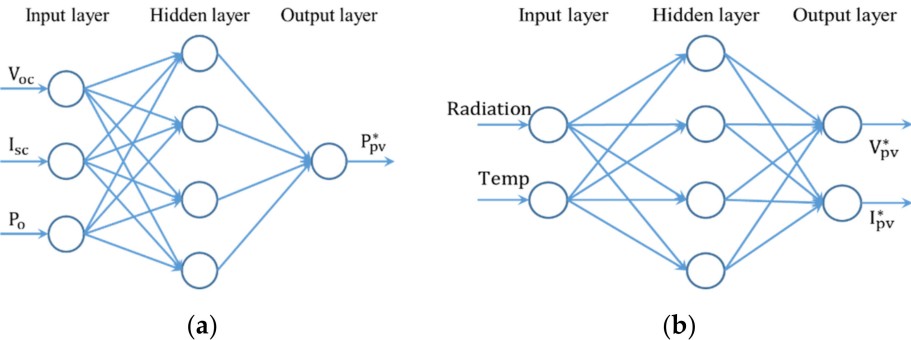

(a)　　　　　　　　　　　　　　　　　(b)

**Figure 16.** *Cont.*

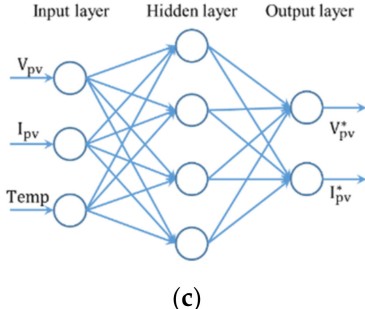

(**c**)

**Figure 16.** Neural network configuration by inputs (**a**) Electrical input; (**b**) Non-electrical input; (**c**) Combined electrical and non-electrical input.

The performance of MPPT control using neural networks depends on the number of nodes and training time. The weight between the nodes is learned by the backpropagation algorithm of the neural network, and the ability and performance to perform MPPT are determined through such learning.

*3.8. OCV and P&O Combination Method*

The P&O method has the disadvantage of being unable to track the actual maximum power point for rapidly changing radiation [89]. The rapidly changing environment of insolation can be measured using a radiation sensor; if the change in power is used without an additional sensor, however, the rapidly changing environment can be observed [90]. Equations (28) and (29) show the radiation change using the change of power.

$$\frac{\Delta P_{pv}}{P_{pv}} < 0.01 : \text{Slow change} \qquad (28)$$

$$\frac{\Delta P_{Pv}}{P_{pv}} > 0.01 : \text{Fast change} \qquad (29)$$

Figure 17 shows the flowchart of the MPPT control method combined with the OCV method and the P&O method [91]. This method detects the rapidly changing conditions of radiation through the change of power, controls the conventional P&O method when the radiation changes slowly, and tracks the maximum power point approximated by the OCV method in the rapidly changing conditions. This compensates for the problem of the P&O method, which cannot track the actual maximum power point in rapidly changing radiation.

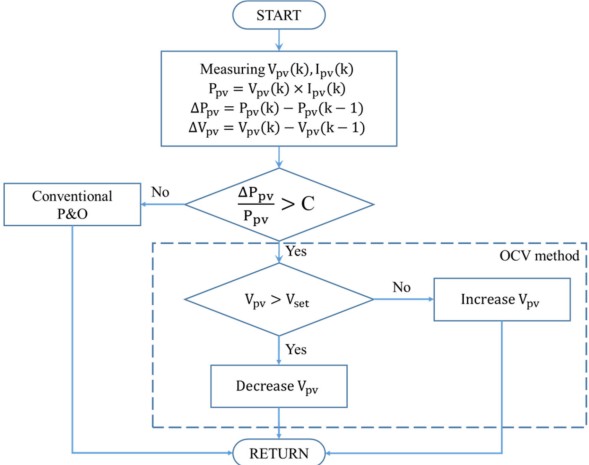

**Figure 17.** Flowchart of the P&O and OCV combination method.

### 3.9. Proportional Integral (PI) Controller and Fuzzy Control Combination Method

The PI controller is a controller that uses two gain values. These two gains are generally fixed, and the response performance varies depending on the size. Since the fixed gain cannot satisfy both transient state and steady state, methods for adjusting the gain value according to the operating state have been proposed. Figure 18 shows how to adjust the gain value of a PI controller using a fuzzy controller [92].

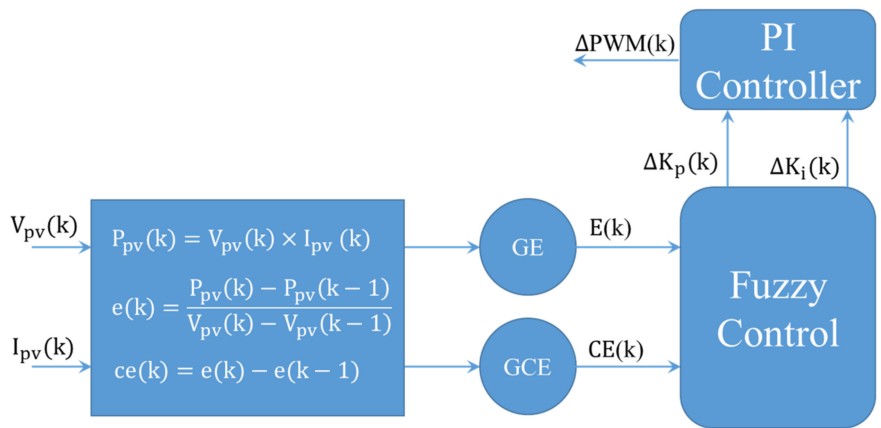

**Figure 18.** PI controller and Fuzzy control combination method.

## 4. Result and Analysis

This paper introduces the configuration, control method, and characteristics of various methods for tracking the maximum power point of the PV system. The MPPT of the PV system is a very important factor for improving power and efficiency. The CV, OCV, and SCC methods introduced in this paper have the advantage of simple algorithm and implementation. Since these methods approximately set a reference value for maximum power point tracking, however, tracking accuracy is low.

The P&O method and the IncCond method continuously measure the current voltage and current of the PV system and use the measured result to track the maximum power point in a predetermined order. Therefore, they are the most commonly used methods for MPPT, because it is possible to track the maximum power point relatively accurately according to environmental change. Nonetheless, there is a problem, i.e., performance is highly dependent on the changing value, which controls the reference value. If this value is large, the tracking speed is fast, but the vibration occurs in steady state, thus resulting in a large loss. If this value is small, the error is reduced in steady state, but the tracking speed is slowed. In order to overcome these disadvantages, various methods of adjusting the changing value according to the operating state have been proposed. Since the P&O and IncCond methods are controlled in a predetermined order, however, they have the disadvantage of gradually increasing or decreasing from the maximum or minimum value during initial operation.

Fuzzy control and neural networks are artificial intelligence control methods with advantages in processing nonlinear systems. Nonetheless, this method relies heavily on the experience of the designer and the user and requires a high-performance controller compared to other methods, which in turn increases the cost of the system. In particular, the neural network requires a controller for the highest performance because it necessitates a lot of computation and sufficient learning time. The various MPPT methods introduced in this paper can be used in combination to overcome their shortcomings by sharing their strengths.

## 5. Conclusions

Solar energy is a very important energy source for the future because it can be predicted among various renewable energy sources and the amount of energy is very large. Therefore, it is very

important to develop the MPPT technique to increase the efficiency and output of the PV system and improve stability.

This paper introduces the following methods for MPPT control of the PV system. The methods include a CV, OCV, and SCC methods that track maximum power points using approximated values, P&O, and IncCond methods, that perform control in a predetermined order, and fuzzy control and neural network that is artificial intelligent control technique. In addition, methods to compensate for shortcomings and improve performance by mixing each method were also introduced.

The method introduced in this paper will help to select the appropriate MPPT method for engineers and researchers who construct smart grids and Micro Grids or use PV systems alone.

**Author Contributions:** Conceptualization, J.-S.K. and J.-C.K.; Data Curation, J.-S.K.; Formal Analysis, J.-S.K.; Funding Acquisition, J.-H.H.; Investigation, J.-S.K. and J.-H.H.; Methodology, J.-H.H.; Project Administration, J.-H.H.; Resources, J.-S.K. and J.-C.K.; Software, J.-S.K., J.-H.H., and J.-C.K.; Supervision, J.-C.K.; Validation, J.-C.K.; Visualization, J.-H.H. and J.-C.K.; Writing—Original Draft, J.-S.K., J.-H.H., and J.-C.K.; Writing—Review & Editing, J.-H.H.; J.-C.K. All authors have read and agree to the published version of the manuscript.

**Funding:** This work was supported by the National Research Foundation of Korea (NRF) grant funded by the Korea government (MSIT) (No. 2017R1C1B5077157). Furthermore, this research was supported by Energy Cloud R&D Program through the National Research Foundation of Korea (NRF) funded by the Ministry of Science, ICT (NRF-2019M3F2A1073385).

**Conflicts of Interest:** The authors declare no conflict of interest.

## Abbreviations

| | |
|---|---|
| MPPT | Maximum Power Point Tracking |
| PV | PhotoVoltaic |
| CV | Constant Voltage |
| OCV | Open Circuit Voltage |
| Wp | Watt peak |
| SCC | Short Circuit Current |
| P&O | Perturbation and Observation |
| OCV | Open Circuit Voltage |
| MPP | Maximum Power Point |
| NB | Negative Big |
| NS | Negative Small |
| ZE | Zero |
| PS | Positive Small |
| PB | Positive Big |
| PI | Proportional Integral |

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
