# Peer review of "Overview of Maximum Power Point Tracking Methods for PV System in Micro Grid"

_electronics, doi:10.3390/electronics9050816_

Round 1
Reviewer 1 Report
The research concept of the review article seems very interesting. The contents are well described. There are a few considerations that must be addressed.
a. The manuscript needs thorough revision to correct few grammatical and spelling mistakes.
b. The format of the equations, mainly the font type and size, needs to be modified.
c. As a technical review, open circuit voltage and short circuit current characteristics plots are expected for different types of PV modules. In fact, in MATLAB/Simulink, IV and PV characteristic plots can be drawn with different temperatures for different PV modules.
d. The References section is very good, yet should be improved. Since the MPPT methods are studied here for microgrid applications, a few papers on PV DC/AC/hybrid microgrids can be cited here. Suggestions are provided here that describe scalable DC microgrids powered by PV arrays and contain MPPT modules. These scalable microgrids ensure MPPT using P & O and incremental conductance methods, respectively. These are employed to support a number of power management units connected to energy storage systems. The referred links are below.
i. https://ieeexplore.ieee.org/document/8636706
ii. https://arxiv.org/abs/1801.00907
iii. https://ieeexplore.ieee.org/document/8636808
iv. https://onlinelibrary.wiley.com/doi/abs/10.1002/tee.23128
Authors can consider more papers on PV microgrids as additional references.
Author Response
Comments and Suggestions for Authors
The research concept of the review article seems very interesting. The contents are well described. There are a few considerations that must be addressed.
- The manuscript needs thorough revision to correct few grammatical and spelling mistakes.
-----
Reply -----
-----
First of all, we are sorry for the inconvenience. If you tell me a paragraph that is not clear and difficult to understand, I will submit it in a clear, easy-to-understand way.
The First author [1], Co Corresponding author [2], Corresponding author [3] are majoring in computer engineering theories for Grid Computing. Although one of the drawbacks of the Grid Computing-related studies is that the descriptions and explanations can be quite lengthy, but their advantage is that the contents of the proposal can be understood clearly just by reading them.
[1] Jae-Sub Ko
https://scholar.google.co.kr/citations?user=Vph8kvsAAAAJ&hl=ko
[2] Jun-Ho Huh
https://scholar.google.co.kr/citations?user=cr5wjNYAAAAJ&hl=ko
[3] Jong-Chan Kim
https://scholar.google.co.kr/citations?user=YGPImkUAAAAJ&hl=ko
At the same time, to cover the drawbacks, the contribution parts have been included in every possible section while correcting the contents with the help of a Native English Speaker to improve readability within a limited time frame. It seems that the special edition ‘Electronic Solutions for Artificial Intelligence Healthcare [4]’, to which we’ve submitted our study, was to provide some understanding to the AI engineers studying computer engineering from the energy point of view. The revised or added parts are being highlighted in red for your possible re-review.
[4] https://www.mdpi.com/journal/electronics/special_issues/AI_Healthcare
- The format of the equations, mainly the font type and size, needs to be modified.
-----
Reply-----
-----
We checked the Equation, font type, and size as a whole by referring to Electronics' template.
- As a technical review, open circuit voltage and short circuit current characteristics plots are expected for different types of PV modules. In fact, in MATLAB/Simulink, IV and PV characteristic plots can be drawn with different temperatures for different PV modules.
-----
Reply-----
-----
We added information about different types of PV modules according to the reviewer's opinion.
ADD 1)
Figure 3 shows the location of the open circuit voltage ( and maximum power point (MPP) voltage ( when rated power of PV module is 110[Wp] and 30[Wp].
Figure 3. Location of and
ADD 2)
In Figure 3, the maximum power point is included in the range of Equation 8, but depending on the value selected, the OCV method can deviate significantly from the actual maximum power point.
ADD 3)
(a)
(b)
Figure 5. P-I curve of PV module
(a) PV module : 110 [Wp]; (b) PV module : 30[Wp]
- The References section is very good, yet should be improved. Since the MPPT methods are studied here for microgrid applications, a few papers on PV DC/AC/hybrid microgrids can be cited here. Suggestions are provided here that describe scalable DC microgrids powered by PV arrays and contain MPPT modules. These scalable microgrids ensure MPPT using P & O and incremental conductance methods, respectively. These are employed to support a number of power management units connected to energy storage systems. The referred links are below.
- https://ieeexplore.ieee.org/document/8636706
- https://arxiv.org/abs/1801.00907
iii. https://ieeexplore.ieee.org/document/8636808
- https://onlinelibrary.wiley.com/doi/abs/10.1002/tee.23128
Authors can consider more papers on PV microgrids as additional references.
-----
Reply-----
-----
We have added reference and related information according to the reviewer's opinion.
ADD 1)
The micro grid is constructed by connecting small-scale power grids to each other, and can replace existing large-scale power generation systems using fossil fuels, and has the advantage of reducing transmission loss because it is produced in places where energy is required. Micro grid consists of DC power grid and distributed power. The PV system is the most representative distributed power supply for micro grids, and MPPT control is used to stably operate multiple power management units through distributed power, and the representative MPPT methods, P & O method [64-66] and IncCond method [67] Is used.
ADD 2)
- A. S. Md. Khalid Hasan, D. Chowdhury and M. Z. Rahman Khan, "Scalable DC Microgrid Architecture with a One-Way Communication Based Control Interface," 2018 10th International Conference on Electrical and Computer Engineering (ICECE), Dhaka, Bangladesh, 2018, pp. 265-268, doi: 10.1109/ICECE.2018.8636706.
- A. S. Md. Khalid Hasan, D. Chowdhury and M. Z. Rahman Khan, “Performance Analysis of a Scalable DC Microgrid Offering Solar Power Based Energy Access and Efficient Control for Domestic Loads,” Electrical Engineering and Systems Science – Signal Processing, 2018
- D. Chowdhury, A. S. Md. Khalid Hasan and M. Z. Rahman Khan, "Scalable DC Microgrid Architecture with Phase Shifted Full Bridge Converter Based Power Management Unit," 2018 10th International Conference on Electrical and Computer Engineering (ICECE), Dhaka, Bangladesh, 2018, pp. 22-25, doi: 10.1109/ICECE.2018.8636808.
- Chowdhury, D., Hasan, A. S. M. K., & Khan, M. Z. R. (2020). Islanded DC Microgrid Architecture with Dual Active Bridge Converter-Based Power Management Units and Time Slot-Based Control Interface. IEEJ Transactions on Electrical and Electronic Engineering. doi:10.1002/tee.23128

Reviewer 2 Report
Dear Authors,
first of all, thank you for submitting your review work for possible publication in Electronics journal. The article deals with a literature review of MPPT control techniques for PV systems in microgrid. The research topic is up-to-date and interesting. Valuable information are reported for each MPPT technique, which could be useful to students, PhD students, engineers, researchers. The manuscript is well written and structured. However, to be fully considered for publication, some issues must be answered and further discussion must be provided. Please find below my remarks and feedback to take into consideration for a second review round process:
1) In the abstract section, the acronym "PV" has to to be defined at the first sentence, and not the second sentence. Besides, it has to be defined also at the beginning of the introduction section.
2) In the introduction section, I suggest you to cite relevant reviews on this topic to emphasize your contributions with this review (it could be useful to provide the reasons for which you carry out this work). Please see below some relevant references to cite:
-M. Seyedmahmoudian, B. Horan, T. Kok Soon, R. Rahmani, A. Muang Than Oo, S. Mekhilef, A. Stojcevski, State of the art artificial intelligence-based MPPT techniques for mitigating partial shading effects on PV systems – A review, Renewable and Sustainable Energy Reviews, Volume 64, 2016, Pages 435-455.
-Gupta, nbspSakshi and nbspNeha sharma. “A Literature Review of Maximum Power Point tracking from a PV array with high Efficiency.” International Journal of Engineering Development and Research 4 (2016): 157-161.
-T. Logeswaran, A. SenthilKumar, A Review of Maximum Power Point Tracking Algorithms for Photovoltaic Systems under Uniform and Non-uniform Irradiances, Energy Procedia, Volume 54, 2014, Pages 228-235.
3) Please add some references in section 2 regarding the PV model.4) The section "MPPT methods" has to be the third and not the second. Please correct it. 5) In my opinion, a discussion about the different methods must be brought as fourth section (before the conclusion) to summarize MPPT algorithms (advantages, drawbacks), key performances, and so on...
Author Response
Dear Authors,
first of all, thank you for submitting your review work for possible publication in Electronics journal. The article deals with a literature review of MPPT control techniques for PV systems in micro grid. The research topic is up-to-date and interesting. Valuable information are reported for each MPPT technique, which could be useful to students, PhD students, engineers, researchers. The manuscript is well written and structured. However, to be fully considered for publication, some issues must be answered and further discussion must be provided. Please find below my remarks and feedback to take into consideration for a second review round process:
1) In the abstract section, the acronym "PV" has to to be defined at the first sentence, and not the second sentence. Besides, it has to be defined also at the beginning of the introduction section.
-----
Reply-----
-----
First of all, we are sorry for the inconvenience. If you tell me a paragraph that is not clear and difficult to understand, I will submit it in a clear, easy-to-understand way.
We revised the relevant parts according to the reviewer's opinion.
The revised or added parts are being highlighted in red for your possible re-review.
CHANGE 1)
Abstract: This paper presents an overview of the maximum power point tracking (MPPT) methods for photovoltaic (PV) systems used in Micro Grids of PV systems.
CHANGE 2)
- Introduction
The photovoltaic (PV) system is receiving much attention because it is an infinite, eco-friendly energy source.
2) In the introduction section, I suggest you to cite relevant reviews on this topic to emphasize your contributions with this review (it could be useful to provide the reasons for which you carry out this work). Please see below some relevant references to cite:
-----
Reply-----
-----
We have added the relevant content according to the reviewer's opinion.
ADD 1)
- Introduction
This paper introduces various methods for MPPT control of the PV system, which is receiving much attention as alternative energy and is most preferred for constructing smart grids and micro grids. It explains the operation principle, structure, advantages, and disadvantages of various MPPT methods, and introduces methods to overcome the disadvantages of each method. The various MPPT methods introduced in this paper will help engineers and researchers using PV systems to select the appropriate MPPT method according to the type, location, and environmental conditions of the PV system. In addition, it is expected that various ideas will be provided to study methods for improving the conventional MPPT method through a method in which the existing MPPT method is mixed with each other.
3) Please add some references in section 2 regarding the PV model.
-----
Reply-----
-----
We have added the relevant content according to the reviewer's opinion.
ADD 1)
- K. Ishaque, Z. Salam, and H. Taheri, “Simple, fast and accurate two diode model for photovoltaic modules,” Solar Energy Mater. Solar Cells, 2011, Vol. 95, pp. 586–594.
- K. Ishaque, Z. Salam, and H. Taheri, “Accurate MATLAB simulink PV system simulator based on a two-diode model,” J. Power Electron., 2011, Vol. 11, pp. 179–187.
- Reisi AR, Moradi MH, Jamasb S. Classification and comparison of maximum power point tracking techniques for photovoltaic system: A review. Renewable and Sustainable Energy Reviews, SciVerse Science Direct, 2013; 433–443.
4) The section "MPPT methods" has to be the third and not the second. Please correct it. 5) In my opinion, a discussion about the different methods must be brought as fourth section (before the conclusion) to summarize MPPT algorithms (advantages, drawbacks), key performances, and so on...
-----
Reply-----
-----
We modified the section number according to the reviewer's opinion, added section 4 result and analysis, and revised section 5 conclusion.
CHANGE 1)
- MPPT methods
3.1. Constant Voltage Method
3.2. Open Circuit Voltage (OCV) Method
3.3. Short Circuit Current (SCC) Method
3.4. Perturbation & Observation (P&O) Method
3.5. Incremental Conductance (IncCond) Method
3.6. Fuzzy MPPT method
3.7. Neural Network Method
3.8. OCV and P&O combination method
3.9. Proportional Integral (PI) controller and Fuzzy control combination method
ADD 1)
- Result and Analysis
This paper introduces the configuration, control method, and characteristics of various methods for tracking the maximum power point of the PV system. The MPPT of the PV system is a very important factor for improving power and efficiency. The CV, OCV, and SCC methods introduced in this paper have the advantage of simple algorithm and implementation. Since these methods approximately set a reference value for maximum power point tracking, however, tracking accuracy is low.
The P&O method and the IncCond method continuously measure the current voltage and current of the PV system and use the measured result to track the maximum power point in a predetermined order. Therefore, it is the most commonly used method for MPPT because it is possible to track the maximum power point relatively accurately according to environmental change. Nonetheless, it has a problem, i.e., performance is highly dependent on the changing value, which controls the reference value. If this value is large, the tracking speed is fast, but the vibration occurs in steady state; thus resulting in a large loss. If this value is small, the error is reduced in steady state, but the tracking speed is slowed. In order to overcome these disadvantages, various methods of adjusting the changing value according to the operating state have been proposed. Since the P&O and IncCond methods are controlled in a predetermined order, however, they have the disadvantage of gradually increasing or decreasing from the maximum or minimum value during initial operation.
Fuzzy control and neural networks are artificial intelligence control methods with advantages in processing nonlinear systems. Nonetheless, this method relies heavily on the experience of the designer and the user and requires a high-performance controller compared to other methods, which in turn increases the cost of the system. In particular, the neural network requires a controller for the highest performance because it necessitates a lot of computation and sufficient learning time. The various MPPT methods introduced in this paper can be used in combination to overcome their shortcomings by sharing their strengths.
CHANGE 2)
- Conclusion
Solar energy is a very important energy source for the future because it can be predicted among various renewable energy sources and the amount of energy is very large. Therefore, it is very important to develop the MPPT technique to increase the efficiency and output of the PV system and improve stability.
This paper introduces the following methods for MPPT control of the PV system. The methods include a CV, OCV, and SCC methods that track maximum power points using approximated values, P&O, and IncCond methods that perform control in a predetermined order, and fuzzy control and neural network that is artificial intelligent control technique. In addition, methods to compensate for shortcomings and improve performance by mixing each method were also introduced.
The method introduced in this paper will help to select the appropriate MPPT method for engineers and researchers who construct smart grids and micro grids or use PV systems alone.

Round 2
Reviewer 1 Report
The revised article addresses the review comments.
Thank you.
Reviewer 2 Report
Dear Authors,
thank you for taking into consideration my previous feedback.
I confirm you the brought replies and the corrected manuscript match my expectation. The manuscript can be accepted in its current form.
Thank you again for submitting this review work in Energies.